# Impact of non-agricultural employment on industrial structural upgrading -Based on the household consumption perspective

Qian Wei, Cao Jian 🔟 *

Department of Economics and Management, Northeast Agricultural University, Harbin, Heilongjiang, China

* 1065096790@qq.com

## Abstract

China's rural revitalization strategy has expanded non-agricultural employment opportunities for rural residents. This has directly raised farmers' incomes and household expenditures, which in turn has contributed to the upgrading of industrial structure. Using provincial data for 2012–2021, our study investigates how this employment transition affects industrial development. The effect of rural residents' consumption expenditures on this relationship is also explored through a linkage model to help measure the extent of the impact. This study further explores the regional differences in this effect and its robustness. The findings suggest that non-farm employment significantly contributes to industrial structural upgrading. However, this effect is not consistent across regions. Moreover, rural residents' consumption plays a pivotal role in this relationship. Governments should therefore encourage more non-farm jobs, stimulate domestic demand and use rural consumption as a key growth catalyst, especially after the demographic dividend disappears. It is also important to take into account regional nuances in policy formulation and make adjustments to cater for these differences to prevent any potential imbalances.

**Data Availability Statement:** All relevant data are within the manuscript and its Supporting Information files.

## I. Introduction

Under the new normal, China's economy has changed from high-speed development to high-quality development, and accelerating the upgrading of industrial structure is still one of the effective means for China's economic development. At present, the proportion of China's primary industry continues to decrease, the proportion of tertiary industry far exceeds that of the secondary industry, and the degree of integration of manufacturing and service industries is deepening, and the layout of the industrial structure still needs to be further improved [1]. A large number of laborers keep flowing from rural to urban areas and from the primary industry to the secondary and tertiary industries, which stimulates the household consumption of rural residents, and the change in the structure of consumer demand triggers consumption upgrading, which becomes an important driving force to lead the industrial upgrading [2]. According to the "Engel effect", consumption upgrading can promote the upgrading and modernization of secondary and tertiary industries, and push forward the process of upgrading industrial structure [3]. Therefore, promoting the increase of non-agricultural employment

**Funding:** The author(s) received no specific funding for this work.

**Competing interests:** The authors have declared that no competing interests exist.

has far-reaching significance for realizing common prosperity and accelerating the transformation of industrial structure, which has gained more and more scholars' attention.

W.A. Lewis (1954) established the first population mobility model, put forward the concept of "surplus labor", pointed out that the transfer of surplus labor from the less productive rural sector to the more productive industrial sector can effectively optimize the allocation of labor resources, and explored the problem of the transfer of surplus labor from rural areas to urban areas, which later gave rise to a series of population mobility theories [4]. It also explored the problem of transferring surplus rural labor to urban areas, which later gave rise to a series of population mobility theories. In the process of China's urbanization, large-scale rural labor migration has become an important social phenomenon. Scholars at home and abroad have not yet reached a consistent conclusion on whether labor mobility between different industries and regions, especially rural labor mobility in the process of urbanization and industrialization, can affect the upgrading of industrial structure. Some researchers believe that inter-regional labor mobility will have a positive impact on industrial structure adjustment (Hanson & Slaughter) [5], promote industrial structure upgrading, and there will also be a certain "reverse" effect (Liu Xinqiang, 2012) [6]. T.W. Shultz discussed the costs and benefits of labor migration on the basis of Lewis in the collection of papers published in 1990 [7]. Based on this, some studies found that due to the limitation of technology level, laborers in rural areas can only engage in low-end manufacturing or service industry after moving to towns, which is not conducive to the upgrading of industrial structure [8]. John. R. Harris and Michael. P. Todaro (1970) further proposed the urban-rural population migration model to explain the population flow between urban and rural areas in the presence of urban unemployment [9]. However, with the adjustment of industrial structure and the implementation of urban-rural strategy in recent years, the education level of rural residents has been raised, and most of them return to their hometowns to start their own businesses by using the technology and management experience learned in the cities, which produces a diffusion effect, driving the enthusiasm of rural residents for non-agricultural employment, and the transfer of entrepreneurial labor force has been increasing day by day, which promotes the improvement of the consumption level of the rural residents, and further promotes the upgrading of the industrial structure.

The "demographic dividend" refers to the fact that a country's working-age population accounts for a larger proportion of the total population, and the dependency ratio is relatively low, which will provide favorable conditions for economic development. Some scholars have introduced the variable of human capital [10] from the perspective of "demographic dividend" to explore the effect of human capital mismatch in the impact of population mobility on industrial structure upgrading. Kuznets believes that the diversification of consumption demand and the advanced structure of consumption can optimize the industrial structure [11]. Previous studies have explored the promotion of non-agricultural employment on household consumption from the perspectives of consumption effect [12] and consumption type [13]. In addition, the impact of other factors such as resource utilization [14–16], climate change [17–19], new crown epidemics [20, 21], and other stakeholders [22, 23] on the industry or economic development is also considered. According to Maslow's hierarchy of needs theory, people's needs are sequentially progressive. The level of disposable income of farmers is the most important factor in determining the level or scale of consumption of residents (Keynes), as people's income increases, the demand for consumption will be raised from only solving the problem of subsistence to the pursuit of high-quality consumption [24]. The increase in non-agricultural employment will inevitably bring about an increase in residents' income, and the increase in income will enhance people's demand for the quality of consumption, China's population base is large, and consumption upgrading will inevitably have an important impact on the upgrading of industrial structure.

In summary, non-agricultural employment will have an important impact on industrial structure upgrading. Previous studies have explored the impact of labor mobility on industrial structure upgrading, and due to the existence of the "demographic dividend", labor mobility can promote industrial structure upgrading through the comparative advantages formed by labor-intensive industries. However, with the disappearance of the "demographic dividend", the structure of the labor force has changed, and household consumption may become an important channel for non-agricultural employment to promote industrial structure upgrading. Therefore, this paper carries out theoretical analysis on the basis of Todaro's demographic model, and explores the impact of increased non-agricultural employment on industrial structure upgrading and the role of consumption as a channel in it from the perspective of household consumption of rural residents. The possible contributions of this paper are: firstly, introducing the perspective of household consumption to explore the influence mechanism of non-agricultural employment on industrial structure upgrading; secondly, adopting the linkage model in testing the mechanism, verifying the mediating effect of rural residents' consumption in promoting industrial structure upgrading by non-agricultural employment and measuring the mediating effect of rural residents' consumption in the promotion of industrial structure upgrading; and lastly, providing a new idea to analyze the impact of non-agricultural employment on industrial structure upgrading in the wake of the disappearance of the "demographic dividend". Finally, it provides a new research idea to analyze whether the channel of non-agricultural employment's influence on industrial structure upgrading has changed in the period after the disappearance of "demographic dividend", and to explore the economic growth effect of labor mobility.

## II. Theoretical analysis and research hypotheses

The upgrading and adjustment of a country's (region's) industrial structure is usually closely related to the country's consumption demand, factor endowment and technological progress, and labor mobility is an important factor that affects a country's (region's) consumption demand, labor supply, human capital accumulation and labor productivity, so labor mobility will inevitably affect the upgrading of industrial structure [25]. According to Todaro model.

Because of the presence of unemployment in cities, rural laborers need to take into account the employment or unemployment rate in cities and also the real income differentials between rural and urban areas when deciding whether to move to work in the urban industrial sector. As long as the net present of urban incomes is expected to be higher than that of rural areas, there will be a surge in labor mobility from rural to urban areas, i.e., an increase in non-agricultural employment.

In the process of industrial structure upgrading, labor mobility is a necessary way to achieve resource allocation, and population mobility can be explained to a certain extent as the transfer process of surplus labor, according to the research of Lu Zhaoguo et al. The development of agriculture can be divided into three phases, the first phase is the stage of unlimited labor supply, in this phase there is a large amount of explicit surplus labor in traditional agriculture with a marginal labor productivity of 0, and it will be transferred to the After the industry, the agricultural production will be surplus, at this time will produce a demographic dividend, and the industry in the absorption of agricultural surplus labor at the same time will absorb agricultural surplus, for the further expansion of industry to provide a large amount of capital. The second stage is the hidden labor transfer stage, after the transfer of visible surplus labor, the marginal productivity of agricultural labor will increase, but the hidden surplus labor still exists and will continue to flow to the industrial sector, which will cause a reduction in total agricultural output, food prices and wages in the industrial sector to rise, and slow down the

rate of absorption of labor in the industrial sector, which will limit the expansion of the industrial sector. The third stage is the stage of agricultural modernization, at this time the surplus labor force has been absorbed by the industrial sector, the production efficiency of agriculture has reached its highest, the wages of industry and agriculture are the same and are determined by the market, the changes in the consumption structure of rural residents will also be closely related to the upgrading of industrial structure [26]. At this time, if the industrial sector wants to absorb the agricultural labor force, it must pay higher wages than those determined by the marginal productivity of labor, and the economic development realizes the transformation from dualistic to monistic, and then realizes the upgrading of industrial structure.

According to the law of Mathieu Clark, as the level of per capita national income increases, the employed population will be transferred from the primary industry firstly with the increase of per capita national income of the whole society, and the employed population will be transferred to the tertiary industry in a large number of cases. This kind of labor transfer from the primary industry to the secondary and tertiary industries, i.e., the increase of non-agricultural employment, will promote the industrial structure upgrading, i.e., the non-agricultural employment will have a positive impact on the industrial structure upgrading, which is specifically manifested in the increase of the index of industrial structure upgrading. This paper puts forward the following hypotheses:

H1: An increase in non-agricultural employment significantly contributes to industrial structural upgrading.

The above measurement method based on industrial structure upgrading analyzes the direct impact of non-agricultural employment on industrial structure upgrading. Further we can explore the role of rural household consumption as a channel of influence from the perspective of consumption, and analyze the possible paths of non-agricultural employment influencing industrial structure upgrading. According to Keynesian consumption theory, the level of household disposable income is the most important factor in determining the level or scale of household consumption of residents. And the inflow of rural labor into the city will bring about the enhancement of rural residents' household consumption as well as the change of rural residents' consumption structure, which will also bring about a certain impact on the development of industrial structure. Therefore, this paper puts forward the following research hypotheses:

H2: Non-agricultural employment indirectly contributes to the upgrading of the industrial structure through the promotion of rural household consumption.

When exploring the impact of rural household consumption upgrading on the upgrading of industrial structure, some studies have shown that there are differences between the consumption levels of urban and rural residents, which provide space for the development of different industries. On the one hand, the continuous upgrading of consumption can drive the upgrading of China's industrial structure and promote the sustained and rapid growth of the national economy [27]. On the other hand, the change of consumption structure precedes the change of industrial structure, and it can be used as the leader of the change of industrial structure, and the industrial structure is based on the change of consumption structure, and the change of consumption structure will change the supply and demand relationship in the market, and it will bring certain influence on the development of industry. Compared with towns, the consumption of rural residents is dominated by basic living consumption [28]. The upgrading of urban and rural residents' consumption will significantly promote the development of the service industry, which, as one of the three major industries in China, will also promote the upgrading of industrial structure to a certain extent [29]. Therefore, the third hypothesis is proposed in this paper:

**Table 1. Breakdown by region.**

| Area | provinces |
|---|---|
| Eastern part | Beijing, Tianjin, Hebei, Shanghai, Jiangsu, Zhejiang, Fujian, Shandong, Guangdong, Hainan |
| Central Region | Shanxi, Anhui, Jiangxi, Henan, Hubei, Hunan |
| Western region | Inner Mongolia, Guangxi, Chongqing, Sichuan, Guizhou, Yunnan, Shaanxi, Gansu, Qinghai, Ningxia, Xinjiang |
| Northeastern region | Liaoning, Jilin, Heilongjiang |

H3: Increased consumption of rural residents positively affects industrial structure upgrading.

## III. Data sources, variable selection and modeling

### 1. Data sources and basic characteristics of the sample

In this paper, the annual data of 30 provinces in China (excluding Hong Kong, Macao, Taiwan and Tibet) from 2012 to 2021 are selected as samples mainly from China Statistical Yearbook and China Urban Statistical Yearbook, in which the annual data involving changes in statistical indicators and calibers are obtained by substituting variables and calculations, and the annual data of some provinces are missing, which are made up by calculating the mean value or the annual growth rate, and the data involving the Raw annual data on import and export trade are converted into RMB using the middle exchange rate of the year.

In addition to the method of dividing the 30 provinces according to the 30 provinces, this paper also divides the 30 provinces into eastern, central, western and northeastern regions to test the heterogeneity of the regions, in which the regions are categorized as shown in Table 1 below:

The descriptive statistics of the variables are shown in Table 2 below, and the time series chart of the index of industrial structure upgrading of the 30 provinces for the years 2012–2021 is shown in Fig 1:

### 2. Variable selection

**(1) Explained variable—industrial structure upgrading.** In this paper, we refer to the practice of Lan Qingxin (2013) [30] to measure industrial structure upgrading and construct the overall industrial structure upgrading index R. Since industrial structure upgrading has a stage, the process of industrial structure upgrading is reflected according to the relative changes in the proportion of output value of the three industries. The first, second and third industries are respectively assigned weights from low to high, and multiplied by the proportion of the output value of each industry in the total output value to obtain the overall industrial

**Table 2. Descriptive statistics of variables.**

| | variable name | sample size | average value | minimum value | maximum values |
|---|---|---|---|---|---|
| implicit variable | Industrial structure upgrading index | 300 | 2.38846 | 2.18 | 2.836 |
| Core independent variables | Level of non-agricultural payrolls | 300 | 0.700702 | 0.3488 | 0.9817 |
| control variable | Infrastructure development level index | 300 | 0.13978 | 0.07 | 0.2077 |
| | Index of the level of government fiscal expenditures | 300 | 0.262591 | 0.105 | 0.7583 |
| | Index of the level of investment in fixed assets | 300 | 0.7689662 | 0.1745 | 1.4312 |
| | Index of the level of openness to the outside world | 300 | 0.1113357 | 0.0001 | 0.8566 |
| intermediary variable | Per capita consumption of rural residents (ten thousand yuan) | 300 | 1.117148 | 0.39017 | 2.72048 |

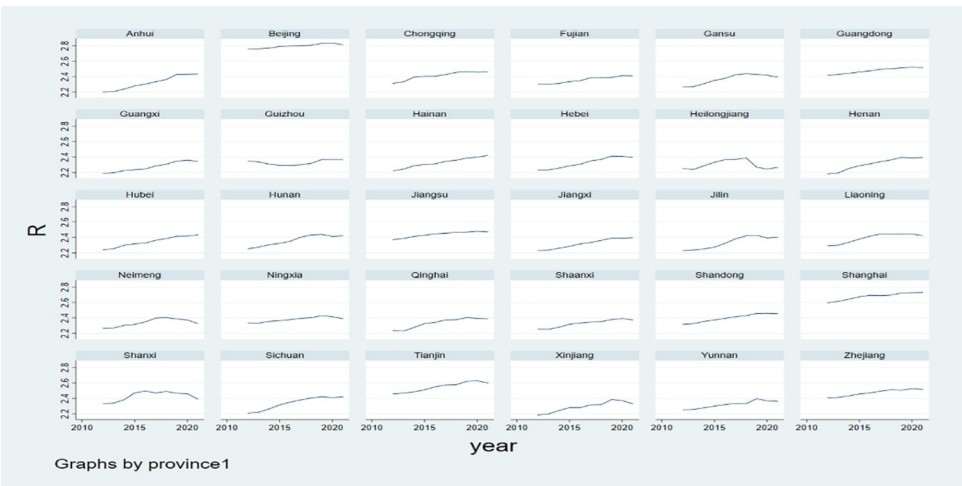

**Fig 1. Time series of industrial structure upgrading index of 30 provinces from 2012 to 2021.**

structure upgrading index R, R = $\sum \sum$ ith industry output value share × i, where the value of i is in the range of [1, 3]. The closer R is to 1, the lower the level of industrial structure development; the closer R is to 3, the higher the level of industrial structure development.

**(2) Core explanatory variable—non-agricultural employment.** This paper takes each province as the research unit, and uses the ratio of employed people engaged in non-agricultural industries in villages to all rural employees in each province from 2012–2021 to measure non-agricultural employment, which is expressed by the rural non-agricultural employment index Innonag. Due to the availability of data, the number of people engaged in non-agricultural industries in the countryside is expressed by the rural population * the proportion of people engaged in secondary and tertiary industries in each province.

Innonag = Rural Employed Population * (Employed Population in Secondary and Tertiary Industries in Each Province / Total Employed Population in Each Province)

**(3) Mediating variable—household consumption.** This paper uses the per capita consumption expenditure of rural households from 2012 to 2021 to reflect the household consumption of residents, which is defined as the "household consumption" variable Income, in ten thousand yuan.

**(4) Control variables.** Referring to the existing research (Long Shaobo and Dingdier, 2022) [31], in order to alleviate the possible problem of omitted variables, the control variables that may affect the upgrading of industrial structure are selected, including the following four control variables:

The level of opening to the outside world (OPEN): the study of Zhao Yunpeng and Ye Jiao (2018) [32] shows that OFDI has a significant negative correlation with the index of rationalization of industrial structure, indicating that OFDI promotes the upgrading of industrial structure with a certain lag effect. This paper adopts the proportion of the total import and export amount of each province to the GDP as a measurement method.

Fiscal expenditure level (fisc): the government's fiscal expenditure means and fiscal policy can promote industrial structure upgrading to a certain extent, which is specifically manifested in the fact that the government adjusts the allocation of resources among industries by adjusting fiscal expenditure, and then promotes industrial structure upgrading. Yang Zhian and Li Menghan (2019) [33] believe that there is a linear correlation between fiscal expenditure policy and industrial structure upgrading. Therefore, this paper incorporates the control variable of

fiscal expenditure level and adopts the ratio of fiscal expenditure to GDP in each province to measure the level of fiscal expenditure.

Infrastructure level (infra): the study by Xu et al. (2021) [34] concludes that scientific assessment of the impact of infrastructure on industrial structure upgrading helps to make differentiated investments and accelerate the pace of industrial structure transformation. Some studies have also confirmed that both new and traditional infrastructure can promote the upgrading of China's industrial structure (Zhao Xiaoqiang and Liu Liping, 2022) [35]. In this paper, the ratio of urban road area to built-up land area in each province is used to measure the level of infrastructure in each province.

Level of fixed asset investment (inve): fixed asset investment promotes industrial structure upgrading through the allocation and flow of allocation factors among industries, so the level of fixed asset investment is also one of the important influencing factors affecting industrial structure upgrading. This paper adopts the proportion of total investment in fixed assets of the whole society to GDP as the measurement index, referring to the article of Long Shaobo et al. [31].

## 3. Basic model setting

**(1) Benchmark regression model.** In order to verify the promotion effect of non-agricultural employment on industrial structure upgrading, this paper constructs the following benchmark regression model:

$$R_{it} = \alpha_0 + \alpha_1 innonag_{it} + \alpha_2 infra_{it} + \alpha_3 fisc_{it} + \alpha_4 inve_{it} + \alpha_5 open_{it} + \varepsilon_{1it} \qquad (1)$$

1. where the subscripts $i$ and $t$ denote provinces and years, respectively, and $R$ denotes the explanatory variable industrial structure upgrading index, and *Innonag* denotes the non-agricultural employment index, the *infra* denotes the level of infrastructure construction, and *fisc* denotes the level of government fiscal expenditure, and *inve* denotes the level of fixed asset investment, *open* denotes the level of opening up to the outside world, and $\varepsilon_{it}$ is the random error term;

**(2) Mediated effects modeling.** In order to test the mediation mechanism of this paper, on the basis of the benchmark regression model, rural residents' consumption is selected as the mediating variable to analyze the impact of non-agricultural employment on industrial structure upgrading, and the following mediation effect model is constructed with reference to He Dongmei and Liu Peng (2020) [36]:

$$Incom_{it} = \beta_0 + \beta_1 Innonag_{it} + \beta_2 infra_{it} + \beta_3 fisc_{it} + \beta_4 inve_{it} + \beta_5 open_{it} + \varepsilon_{2it} \qquad (2)$$

$$R_{it} = \lambda_0 + \lambda_1 Innonag_{it} + \lambda_2 Incom_{it} + \lambda_3 infra_{it} + \lambda_4 fisc_{it} + \lambda_5 inve_{it} + \lambda_6 open_{it} + \varepsilon_{3it} \qquad (3)$$

In model (2), the subscripts $i$ and $t$ denote provinces and years, respectively, and $R$ denotes the explanatory variable industrial structure upgrading index, and *Innonag* denotes the non-agricultural employment index, and *Incom* denotes rural residents' consumption, and $\varepsilon_{it}$ is the random error term; $\alpha_1$ denotes the total effect of non-agricultural employment rate on industrial structure upgrading, and $\lambda_1$ denotes the direct effect of non-agricultural employment on industrial structure upgrading, and $\beta_1 \times \lambda_2$ denotes the mediating effect of rural residents' consumption. Need to test whether the intermediary effect of rural residents' consumption exists, referring to the method of Wen Zhonglin (2004), the intermediary effect test procedure is as

follows: the first step examines the impact of non-agricultural employment on industrial structure upgrading in Eq (1), and tests whether the coefficient is significant. $\alpha_1$ is significant or not. In the second step, if the coefficient $\alpha_1$ is significant, then test the coefficients in Eq (2) $\beta_1$ and in Eq (3) $\lambda_1$ whether it is significant. In the third step, if the coefficients of $\beta_1$ and $\lambda_2$ are both significant and $\lambda_1$ significant, it indicates that there is a partial mediation effect; if $\beta_1$ and $\lambda_2$ are both significant but $\lambda_1$ are not significant, it indicates a full mediation effect. In the fourth step, if $\beta_1$ and $\lambda_2$ at least one is not significant, then a Sobel test is required to determine whether a mediating effect exists [37].

## IV. Empirical analysis and results

### 1. Benchmark regression results and analysis at the national level

Table 3 presents the regression results based on the benchmark model. Among them, regression (1) represents the regression results when no control variables are added, and regression (2) represents the regression results when control variables are added. The results of regression (1) show that the level of non-agricultural employment is significant at the 1% level, and for every 1% increase in the level of non-agricultural employment, the index of industrial structure upgrading is increased by 0.7277%; after adding the control variables, the level of non-agricultural employment is still significant at the 1% level, and the coefficient is positive. This indicates that non-agricultural employment enhancement significantly promotes industrial structure upgrading, which in turn initially confirms the establishment of hypothesis H1, i.e., the increase of non-agricultural employment will promote industrial structure upgrading. Specific reasons this paper argues that the increase in non-agricultural employment has led to the flow of the employed agricultural population to the secondary and tertiary industries, promoted the increase in output value of the secondary and tertiary industries, and is conducive to the promotion of the transfer of the primary industry to the secondary and tertiary industries in rural areas.

**Table 3. Plot of benchmark regression results.**

|  | R | |
|---|---|---|
|  | **Return (1)** | **Returns (2)** |
| **Innonag** | 0.7277*** | 0.7551*** |
|  | (-21.15) | (-16.95) |
| **Infra** | - | -0.4618*** |
|  | - | (-2.59) |
| **Fisc** | - | 0.2750*** |
|  | - | (-5.46) |
| **Inve** | - | -0.1285*** |
|  | - | (-5.69) |
| **Open** | - | -0.0388 |
|  | - | (-1) |
| **ratio** | 1.8786*** | 1.9549*** |
|  | (-76.53) | (-47.9) |
| **sample size** | 300 | 300 |
| **Adj-R$^2$** | 0.7543 | 0.6747 |

Note: ①

***, **, and * indicate 1%, 5%, and 10% significance levels, respectively; ② t-values are in parentheses.

In terms of control variables, the level of government fiscal expenditure (Fisc) is significant and the coefficient is positive, indicating that the government's fiscal expenditure has a significant role in promoting industrial structure upgrading, i.e., the government is able to promote the upgrading of industrial structure; the level of infrastructure construction (Infra) is significant and the coefficient is negative, indicating that an increase in the level of infrastructure construction inhibits the upgrading of industrial structure; the level of fixed asset investment (Inve) is significant and the coefficient is negative, indicating that the increase of fixed asset investment will significantly inhibit the upgrading of industrial structure, which also verifies the research of Hong Jiao (2020) that fixed asset investment has a reverse inhibition effect on the upgrading of industrial structure [38]; the coefficient of the level of opening up to the outside world (Open) is not significant, which is not considered to have a significant effect on the upgrading of industrial structure. The consistency of the direction of the coefficients of the above control variables with reality indicates that the empirical results of this paper are more reliable.

## 2. Baseline regression results and analysis at the district level

In this paper, the whole sample is divided into the east, center, west and northeast for regression respectively, aiming at exploring the regional differences in the impact of non-agricultural employment on industrial structure upgrading, and the estimation results are shown in Table 4:

First of all, non-agricultural employment in the eastern, central and western regions all has a significant positive impact on industrial structure upgrading, indicating that non-agricultural employment can promote industrial structure upgrading in most regions of the country, and that the increase in non-agricultural employment enhances the mobility of rural laborers among industries and promotes the rational allocation of laborers among industries, whereas in the northeastern region, since agriculture and heavy industry are dominant, farmers

**Table 4. Map of the results of the regional heterogeneity test.**

|  | R | | | |
|---|---|---|---|---|
|  | **the east** | **central section** | **western part** | **north-eastern** |
|  | **Returns (3)** | **Returns (4)** | **Returns (5)** | **Returns (6)** |
| **innonag** | 1.0320*** | 0.4422*** | 0.2286** | 0.6419 |
|  | (15.48) | (2.74) | (2.44) | (1.33) |
| **infra** | -0.5988*** | 1.4060** | 0.7269** | 0.0209 |
|  | (-2.00) | (2.40) | (2.39) | (0.02) |
| **fisc** | 0.6778*** | 0.3422 | 0.0638 | 0.3731 |
|  | (6.38) | (1.39) | (0.94) | (0.83) |
| **inve** | -0.1986*** | 0.0364 | 0.0026 | -0.1037* |
|  | (-5.21) | (0.69) | (0.08) | (-1.85) |
| **open** | -0.1644*** | -0.8936* | 0.2164 | 0.3474 |
|  | (-4.47) | (-1.69) | (1.31) | (0.69) |
| **ratio** | 1.7413*** | 1.7692*** | 2.0708*** | 1.8717*** |
|  | (20.20) | (18.14) | (47.40) | (5.94) |
| **sample size** | 100 | 60 | 110 | 30 |
| **Adj-R²** | 0.8592 | 0.4411 | 0.2815 | 0.2165 |

Note: ①

***, **, and * indicate 1%, 5%, and 10% significance levels, respectively; ② t-values are in parentheses.

engaged in other industries will have a negative spillover effect, so the impact of non-agricultural employment is less significant. Secondly, there are regional differences in the promotion effect of non-agricultural employment on industrial structure upgrading, in which the impact of non-agricultural employment on industrial structure upgrading is the largest in the eastern region, followed by the central region and the smallest in the western region. The reason may be: the eastern region is more economically developed, the rural labor force flows more smoothly among various industries, and non-agricultural employment is more popular, so the degree of industrial structure upgrading caused by non-agricultural employment is larger; while in the central region, due to the richer resources, there is a larger number of rural labor force staying in the countryside, and the increase of non-agricultural employment will also promote the industrial structure upgrading to a certain extent; in the western region, due to the less developed economy, the non-agricultural employment of agricultural households has the greatest impact on the industrial structure upgrading, followed by the smallest in the western part of the country. underdeveloped, the awareness of non-agricultural employment among agricultural households is insufficient, and the index of non-agricultural employment is low, so the impact of the increase in non-agricultural employment on the upgrading of industrial structure is smaller.

### 3. Endogeneity test and robustness test

**(1) Endogeneity test.**   Non-agricultural employment in the promotion of industrial structure upgrading at the same time, industrial structure upgrading will also have a facilitating effect on the increase in non-agricultural employment, the two are mutually causal, mutual influence, so the need to solve the problem of endogeneity. This paper tries to solve the endogeneity problem by using the non-agricultural employment index with one period lag, mainly based on the following considerations: the increase of non-agricultural employment will not have an immediate impact on industrial structure upgrading, and its impact has a certain degree of lag, which will be reflected in the industrial structure upgrading index of the next period, so we consider the use of non-agricultural employment index with one period lag as an explanatory variable for the baseline regression, and the results of the regression show that, after a period lag, its impact on non-agricultural employment will be more and more significant. The regression results show that the impact of non-agricultural employment on industrial structure upgrading is still significant and the coefficient is positive after one period of lagging, indicating that the empirical results of this paper are more reliable and consistent with reality. The results of benchmark regression are shown in Table 5 below.

**(2) Robustness test.**   To ensure the robustness of the conclusions, this paper conducts a robustness test by replacing the variables, including the variable of urbanization (urban) to replace the main explanatory variable of non-agricultural employment, measured by the proportion of urban population in the total population of each province at the end of the year, and after replacing the explanatory variable of urbanization with urbanization, taking into account the endogeneity problem, we also lagged the urbanization by one period, and then re-run the regression analysis. The results show that urbanization also has a significant and positive impact on industrial structure upgrading, supporting the previous conclusion. The results of the robustness test are shown in Table 6.

## V. Mechanism testing

On the basis of the model setting in reference to related literature, due to the correlation between the equation variables, the joint equation model is selected to carry out the mechanism test, and when using the single equation estimation method, the link between the

**Table 5. Baseline regression results with one period lag for explanatory variables.**

|  | R | |
| --- | --- | --- |
|  | **Returns (7)** | **Returns (8)** |
| **Innonag.₋₁** | 0.7072*** | 0.6881*** |
|  | (20.22) | (15.16) |
| **Infra** | - | -0.4169** |
|  | - | (-2.33) |
| **Fisc** | - | 0.2883*** |
|  | - | (5.61) |
| **Inve** | - | -0.1441*** |
|  | - | (-6.31) |
| **Open** | - | -0.0216 |
|  | - | (-0.52) |
| **ratio** | 1.9069*** | 2.0184*** |
|  | (77.08) | (47.29) |
| **sample size** | 270 | 270 |
| **Adj-R$^2$** | 0.6025 | 0.6899 |

Note: ①

***, **, and * indicate 1%, 5%, and 10% significance levels, respectively; ② t-values are in parentheses.

equations will be ignored, so all equations can be estimated as a whole (i.e., system estimation method) is more efficient, this paper adopts the "three-stage least squares method (3SLS)" to carry out the estimation. In this paper, the "three-stage least squares (3SLS)" method is used for estimation. The results of the three-stage least squares method are shown in Table 7 below, in which the results of OLS and Two_sls are used as references.

As can be seen from the table, the level of non-agricultural employment has a significant positive impact on industrial structure upgrading, and for every 1% increase in the level of

**Table 6. Benchmark regression results with replacement of explanatory variables.**

|  | R | |
| --- | --- | --- |
|  | **Returns (9)** | **Returns (10)** |
| **Urban.L1** | 0.8796*** | 0.8808*** |
|  | (26.81) | (19.27) |
| **Infra** | - | -0.3142** |
|  | - | (-2.02) |
| **Fisc** | - | 0.1025** |
|  | - | (2.27) |
| **Inve** | - | -0.1028*** |
|  | - | (-5.01) |
| **Open** | - | -0.1107*** |
|  | - | (-2.93) |
| **ratio** | 1.8731*** | 1.9823*** |
|  | (93.87) | (53.40) |
| **sample size** | 270 | 270 |
| **Adj-R$^2$** | 0.7274 | 0.7589 |

Note: ①

***, **, and * indicate 1%, 5%, and 10% significance levels, respectively; ② t-values are in parentheses.

**Table 7. Plot of 3SLS results.**

| R | OLS | Two_SLS | Three_sls |
|---|---|---|---|
| Innonag | 0.4234*** | 0.6801*** | 0.6801*** |
| | (0.0490) | (0.1180) | (0.1180) |
| Incom | 0.1272*** | 0.0199 | 0.0199 |
| | (0.0158) | (0.0472) | (0.0472) |
| Incom | OLS | Two_SLS | Three_sls |
| Innonag | 2.6878*** | 2.6878*** | 2.6608*** |
| | (0.1549) | (0.1549) | (0.1481) |
| Infra | 2.0897*** | 2.0897*** | 2.7215*** |
| | (0.6200) | (0.6200) | (0.5777) |
| Fisc | 0.4497** | 0.4497** | 0.1158 |
| | (0.1752) | (0.1752) | (0.1617) |
| Inve | -0.3035*** | -0.3035*** | -0.1497** |
| | (0.0785) | (0.0785) | (0.0737) |
| Open | -0.6256*** | -0.6256*** | -0.5925*** |
| | (0.1356) | (0.1356) | (0.1288) |
| obs | 300 | 300 | 300 |

Note: ①

***, **, and * indicate 1%, 5%, and 10% significance levels, respectively; ② t-values are in parentheses.

non-agricultural employment, the level of industrial structure upgrading rises by 0.6801%. It should be noted that in the ordinary least squares regression, there is also a significant positive impact of residents' consumption on non-agricultural employment, and for every 1% increase in residents' consumption, the level of industrial structure upgrading rises by 0.1272%, which suggests that there may be an intermediary effect of residents' consumption, and thus further intermediary effect tests are needed. The lagged one-period variable of non-agricultural employment is used here to carry out the test, which circumvents the problem of endogeneity as much as possible. The test results are summarized in Table 8 below:

As can be seen from the test results in Table 7, non-agricultural employment will have a positive direct effect on industrial structure upgrading on the one hand, and on the other

**Table 8. Plot of the results of the mediation effect test.**

| control variable | there are | not have |
|---|---|---|
| $\alpha_1$ | 0.7072*** | 0.6881*** |
| | (0.0349) | (0.0454) |
| $\beta_1$ | 2.3363*** | 2.4296*** |
| | (0.1105) | (0.1515) |
| $\lambda_2$ | 0.1219*** | 0.1289*** |
| | (0.0179) | (0.0167) |
| $\lambda_1$ | 0.4225*** | 0.3748*** |
| | (0.0528) | (0.0577) |
| intermediary effect | 40.27% | 45.51% |
| direct effect | 59.73% | 54.49% |

Note: ①

***, **, and * indicate 1%, 5%, and 10% significance levels, respectively; ② t-values are in parentheses.

hand, it will also act indirectly on the consumption level of the residents, which in turn will have a positive mediating effect. Specifically, when no control variables are added, the share of mediation effect is 40.27%, while after adding control variables, the share of mediation effect is raised to 45.51%.

## VI. Conclusions, research limitations and policy implications

### 1. Main conclusions

This paper verifies the non-agricultural employment on the promotion of industrial structure upgrading, and tested the different regions acting on the non-agricultural employment on the industrial structure upgrading of the impact of the different regions as well as rural household consumption in non-agricultural employment to promote the process of industrial structure upgrading of the inherent transmission mechanism. The results of this paper show that, firstly, the increase in the level of non-farm employment promotes the upgrading of industrial structure. The level of non-farm employment in the eastern, central and western regions has a significant role in promoting the upgrading of industrial structure, in which the level of non-farm employment in the northeastern region does not have a significant impact on the upgrading of industrial structure. Secondly, in the process of promoting industrial structure upgrading by non-farm employment, there is a partial mediating effect of residents' consumption, and the proportion of the mediating effect increases from 40.27% to 45.51% after adding control variables. Finally, the level of government fiscal expenditure has a significant positive effect on industrial structure upgrading, while among the other control variables, infrastructure construction and fixed asset investment have a significant negative effect on industrial structure upgrading, and the level of opening up to the outside world has no significant effect on industrial structure upgrading.

### 2. Research limitations

Based on the above research, in writing by their own cognitive level and understanding of the constraints of the ability, this paper in the research process there are some limitations: First, the research method: this paper mainly takes the method of empirical analysis, the selection of samples is narrower, the sample size is insufficient, and the data collected are subjective. Second, the content of the study, mainly focusing on the research with the theoretical aspects of non-farm employment and industrial structure upgrading, and the combination of theory and practice is less. Third, the influencing factors of industrial structure upgrading are not considered comprehensively, and the variables are not selected properly.

### 3. Policy implications

In order to better promote the rural revitalization strategy and to achieve the goal of common prosperity as soon as possible, the following policy recommendations are made in accordance with the findings of the study:

First, based on the differences in the impact of the level of non-agricultural employment on the upgrading of industrial structure in different regions, the Government needs to formulate relevant regional policies in accordance with local conditions, to provide development opportunities with precise policies, to implement differentiated regional policy arrangements, to increase the employment opportunities for farmers, and to give better play to the facilitating effect of non-agricultural employment on the upgrading of industrial structure.

Secondly, we should firmly implement the strategy of expanding domestic demand, adjust the consumption structure of the population, raise the income level of the population, improve

the policy of promoting consumption by the population, improve the policy of restricting consumption by the population, adjust the mechanism of income distribution, narrow the income gap, so as to enable more consumption by the population to shift from basic subsistence consumption to enjoyment consumption, and actively guide the consumption of the population to play a better intermediary role in the upgrading of the industrial structure through the promotion of non-agricultural employment. actively guide residents' consumption to play a better intermediary role in promoting the upgrading of the industrial structure, and promote the optimization and upgrading of the industrial structure from the primary to the secondary and tertiary sectors.

Finally, government financial expenditures also have a significant role in promoting the upgrading of industrial structure. The government can appropriately adjust its fiscal policy, improve the efficiency of the use of financial funds, and promote the rational and effective allocation and utilization efficiency of resources, while also rationally arranging its financial expenditures in response to the comparative advantages of various regions and promoting the coordinated development of various industrial sectors in order to realize the upgrading of industrial structure.

## Supporting information

**S1 Data.**
(XLSX)

## Author Contributions

**Conceptualization:** Qian Wei.

**Data curation:** Cao Jian.

**Formal analysis:** Qian Wei.

**Methodology:** Qian Wei.

**Software:** Cao Jian.

**Writing – original draft:** Cao Jian.

**Writing – review & editing:** Cao Jian.

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
