## [Decision Letter · Decision Letter 0]

19 Oct 2023

PONE-D-23-25181Impact of non-agricultural employment on industrial structural upgrading -Based on the household consumption perspectivePLOS ONE

Dear Dr. Cao,

Thank you for submitting your manuscript to PLOS ONE. After careful consideration, we feel that it has merit but does not fully meet PLOS ONE’s publication criteria as it currently stands. Therefore, we invite you to submit a revised version of the manuscript that addresses the points raised during the review process.

We look forward to receiving your revised manuscript.

Kind regards,

Grigorios L. Kyriakopoulos, 2 PhDs, 3 MSc, 2 MA, MEng, 2 BA, BSc

Academic Editor

PLOS ONE

Journal Requirements:

Reviewers' comments:

Reviewer's Responses to Questions

**Comments to the Author**

1. Is the manuscript technically sound, and do the data support the conclusions?

Reviewer #1: Yes

Reviewer #2: Yes

2. Has the statistical analysis been performed appropriately and rigorously? 

Reviewer #1: Yes

Reviewer #2: Yes

3. Have the authors made all data underlying the findings in their manuscript fully available?

Reviewer #1: Yes

Reviewer #2: Yes

4. Is the manuscript presented in an intelligible fashion and written in standard English?

Reviewer #1: Yes

Reviewer #2: Yes

5. Review Comments to the Author

Reviewer #1: The present paper is highly interesting and is a major contribution to scientific

progress. The paper deals with an interesting topic, especially for impact of non-

agricultural employment on industrial structural upgrading. This paper represents a

good attempt, which provides very good insight regarding with the non-agricultural

employment, useful to the readers of journal PLOS ONE. This paper is well

organised. It has a good structure and provides an easy and meaningful reading. The

English writing is acceptable.

I recommend the publication of the paper under the following major revisions:

1. It is necessary to be included further references .

e.g.

a) Zafeiriou, E., Spinthiropoulos, K., Tsanaktsidis, C., Garefalakis, S., Panitsidis, K..

Garefalakis, A. and Arabatzis, G. (2022). “ Energy and Mineral Resources

Exploitation in the Delignitization Era: The Case of Greek Peripheries ”. Energies ,

2022, 15(13), 4732.

b) Arabatzis, G., Tsiantikoudis, S., Drakaki N and Andreopoulou, Z. (2011).

"The LEADER + Community Initiative and the Local Action Groups in Greece".

Journal of Environmental Protection and Ecology, 12 (4A): 2255–2260.

c) Sofios, S., Arabatzis, G and Baltas, V. (2008). "Policy for management of water

resources in Greece". The Environmentalist, 28 (3):185-194.

3. The discussion should be improved. There needs to be more comparative analysis

with other studies.

4. Conclusions should be improved in a more concisely way and compared with

similar studies.

Reviewer #2: The authors deserve praise for presenting a compelling paper that offers substantial contributions to the field. The manuscript stands strong in numerous aspects, but there are certain areas that warrant refinement in future iterations.

1. One noticeable concern is the prolonged sentences scattered throughout the paper. These can make comprehension somewhat daunting. As an illustration, the abstract can be rephrased as:

“China's rural revitalization strategy has expanded non-agricultural job opportunities for rural residents. This job transition directly enhances farmers' incomes and household spending, which in turn advances the industrial structure. Using 2012-2021 provincial data, our study investigates how this job transition impacts industrial progress. Additionally, it probes into the influence of rural household spending on this relationship through a linkage model, helping to gauge the scope of the impact. The study further explores the regional variations of this effect and its robustness. The findings indicate that non-agricultural jobs significantly foster industrial advancements. However, this effect is not uniform across regions. Moreover, rural spending plays a pivotal role in this relationship. As a result, the government should encourage more non-agricultural jobs, stimulate domestic demand, and harness rural consumption as a key growth catalyst, especially after the "demographic dividend" wanes. It's also essential that policies are crafted with regional nuances in mind, making adjustments to cater to these differences and prevent any potential imbalances.”

2. The manuscript could benefit from a section detailing the research's limitations, any uncertainty in the results, and the future work designed to address these gaps.

3. Moreover, in the current global context, the omission of any mention of climate change—a pressing concern—is glaring. The authors might want to incorporate this theme into their analysis. For guidance on this topic, consider referring to the works of:

• Kyriakopoulos, G. L., & Sebos, I. (2023). Enhancing Climate Neutrality and Resilience through Coordinated Climate Action: Review of the Synergies between Mitigation and Adaptation Actions. Climate, 11(5), 105. https://doi.org/10.3390/cli11050105

• Kyriakopoulos, G. L., Sebos, I., Triantafyllou, E., Stamopoulos, D., & Dimas, P. (2023). Benefits and Synergies in Addressing Climate Change via the Implementation of the Common Agricultural Policy in Greece. Applied Sciences, 13(4), 2216. https://doi.org/10.3390/app13042216

• Ioannis Sebos and Leonidas Kallinikos, Greenhouse gas emissions in Greek agriculture: Trends and projections, E3S Web Conf., 436 (2023) 02007, https://doi.org/10.1051/e3sconf/202343602007

4. Given the profound repercussions of the COVID-19 pandemic, its absence from the discussion is notable. To incorporate insights on this, the authors might want to consult papers such as:

• Papadogiannaki, S., Liora, N., Parliari, D., Cheristanidis, S., Poupkou, A., Sebos, I., ... & Melas, D. (2023). Evaluating the Impact of COVID-19 on the Carbon Footprint of Two Research Projects: A Comparative Analysis. Atmosphere, 14(9), 1365. https://doi.org/10.3390/atmos14091365

• Progiou, A. G., Sebos, I., Zarogianni, A. M., Tsilibari, E. M., Adamopoulos, A. D., & Varelidis, P. (2022). Impact of covid-19 pandemic on air pollution: the case of athens, greece. Environmental Engineering and Management Journal, 21(5), 879-889.

5. Inclusion of a stakeholder analysis could further enhance the paper's depth. Potential references on this subject include studies by:

• Ioanna, N., Pipina, K., Despina, C. et al. Stakeholder mapping and analysis for climate change adaptation in Greece. Euro-Mediterr J Environ Integr 7, 339–346 (2022). https://doi.org/10.1007/s41207-022-00317-3

• Sebos, I., Nydrioti, I., Katsiardi, P. et al. Stakeholder perceptions on climate change impacts and adaptation actions in Greece. Euro-Mediterr J Environ Integr (2023). https://doi.org/10.1007/s41207-023-00396-w

6. PLOS authors have the option to publish the peer review history of their article (what does this mean?). If published, this will include your full peer review and any attached files.

Reviewer #1: No

Reviewer #2: **Yes: **Ioannis Sebos

---

## [Author Response · Author response to Decision Letter 0]

25 Oct 2023

Dear Editors and Reviewers:

Thank you very much for the opportunity to revise our manuscript. We appreciate the comments and noting of the significance of this work from the Reviewers and the Editor. Those comments are valuable and helpful for revising and improving our paper, as well as the important guiding significance to our researches. We have revised our manuscript according to these comments and please find our point-by-point replies to Editors and Reviewers comments. The main corrections in the paper and the responds to the reviewer’s comments are as following:

Responds to the reviewer’s comments:

Reviewer #1:

1.Response to comment:“It is necessary to be included further references .”

Response: We sincerely appreciate the valuable comments. We have checked the literature carefully and added more references into the INTRODUCTION part in the revised manuscript.

2.We didn’t see comment with serial number 2 in the email we received.

3. Response to comment:“The discussion should be improved. There needs to be more comparative analysis with other studies.”

Response: We think this is an excellent suggestion. We have analyzed and compared the existing research on human capital and industrial upgrading in lines 32-34 on the third page. However, there is less research in this aspect at present, and it will be supplemented in time in subsequent research.

4.Response to comment: “Conclusions should be improved in a more concisely way and compared with similar studies.”

Response: Thanks for your suggestion. We have revised the research conclusion to a more concise expression.

Reviewer #2:

1.Response to comment:“ One noticeable concern is the prolonged sentences scattered throughout the paper. These can make comprehension somewhat daunting. ”

Response: We tried our best to improve the manuscript and made some changes to the manuscript. Here we did not list the changes but marked in red in the revised paper. We appreciate for Reviewer’s warm work earnestly and hope that the correction will meet with approval.

2.Response to comment:” The manuscript could benefit from a section detailing the research's limitations, any uncertainty in the results, and the future work designed to address these gaps.”

Response: Thanks for your suggestion. We have added the limitations of existing research and prospects for future research. 

3&4.Response to comment “Moreover, in the current global context, the omission of any mention of climate change—a pressing concern—is glaring.” and “Given the profound repercussions of the COVID-19 pandemic, its absence from the discussion is notable.”

Response: As suggested by the reviewer, we have added more references to support our research. We have checked the literature carefully and added more references on climate change and COVID-19 into the INTRODUCTION part in the revised manuscript.

All revisions have been made with respect to the feedback received, and I hope they meet your expectation. Once again, thank you for your constructive critique. In conclusion, I hope the revised manuscript will be deemed suitable for publication. I eagerly await your further comments or decision.

Sincerely,

Jian Cao

---

## [Decision Letter · Decision Letter 1]

30 Oct 2023

Impact of non-agricultural employment on industrial structural upgrading -Based on the household consumption perspective

PONE-D-23-25181R1

Dear Dr. Cao,

We’re pleased to inform you that your manuscript has been judged scientifically suitable for publication and will be formally accepted for publication once it meets all outstanding technical requirements.

Kind regards,

Grigorios L. Kyriakopoulos, 2 PhDs, 3 MSc, 2 MA, MEng, 2 BA, BSc

Academic Editor

PLOS ONE

Additional Editor Comments (optional):

Reviewers' comments:

Reviewer's Responses to Questions

**Comments to the Author**

1. If the authors have adequately addressed your comments raised in a previous round of review and you feel that this manuscript is now acceptable for publication, you may indicate that here to bypass the “Comments to the Author” section, enter your conflict of interest statement in the “Confidential to Editor” section, and submit your "Accept" recommendation.

Reviewer #1: All comments have been addressed

Reviewer #2: All comments have been addressed

2. Is the manuscript technically sound, and do the data support the conclusions?

Reviewer #1: Yes

Reviewer #2: Yes

3. Has the statistical analysis been performed appropriately and rigorously? 

Reviewer #1: Yes

Reviewer #2: Yes

4. Have the authors made all data underlying the findings in their manuscript fully available?

Reviewer #1: Yes

Reviewer #2: Yes

5. Is the manuscript presented in an intelligible fashion and written in standard English?

Reviewer #1: Yes

Reviewer #2: Yes

6. Review Comments to the Author

Reviewer #1: none

Reviewer #2: The authors successfully addressed my comments. I suggest publication. Language has improved. Stakeholder analysis, climate change, COVID-19 effects, etc were reflected in the manuscript.

7. PLOS authors have the option to publish the peer review history of their article (what does this mean?). If published, this will include your full peer review and any attached files.

Reviewer #1: No

Reviewer #2: **Yes: **Ioannis Sempos

---

## [Editor Report · Acceptance letter]

3 Nov 2023

PONE-D-23-25181R1 

Impact of non-agricultural employment on industrial structural upgrading
-Based on the household consumption perspective 

Dear Dr. Jian:

I'm pleased to inform you that your manuscript has been deemed suitable for publication in PLOS ONE. Congratulations! Your manuscript is now with our production department. 

Kind regards, 

on behalf of

Dr. Grigorios L. Kyriakopoulos 

Academic Editor

PLOS ONE